# Evaluation of Oxygen Absorbers Using Food Simulants and Inductively Coupled Mass Spectrometry

**DOI:** 10.3390/foods12193686

**Published:** 2023-10-07

**Authors:** Seung-Yeon Oh, Eun-Ji Kang, Kyung-Jik Lim, Yoon-Hee Lee, Han-Seung Shin

**Affiliations:** Department of Food Science and Biotechnology, Dongguk University-Seoul, 32, Dongguk-ro, Ilsandong-gu, Goyang-si 10326, Gyeonggi-do, Republic of Korea; bagb4305@gmail.com (S.-Y.O.); eunkong127@gmail.com (E.-J.K.); kyung9209@naver.com (K.-J.L.); dldbsgml491@naver.com (Y.-H.L.)

**Keywords:** smart packaging, oxygen absorber, food simulants, inductively coupled plasma-mass spectrometry (ICP-MS), heavy metal

## Abstract

In this study, we developed and validated an analytical method to evaluate the heavy metal elution from an active packaging material’s oxygen absorber to a food simulant. Using water, 4% acetic acid, n-heptane, 20% ethanol, and 50% ethanol as food simulants, we quantified cobalt, copper, platinum, and iron with inductively coupled plasma-mass spectrometry. The method was thoroughly validated for linearity, accuracy, precision, LOD, and LOQ through inter-day and intra-day analysis repetitions. R^2^ values ranged from 0.9986 to 1.0000, indicating excellent linearity. The LOD values ranged from 0.00002 to 0.2190 mg/kg, and the LOQ values ranged from 0.00007 to 0.6636 mg/kg. The method’s accuracy was 95.14% to 101.98%, with the precision ranging from 0.58% to 10.37%. Our results confirmed the method’s compliance with CODEX standards. Monitoring the oxygen absorber revealed undissolved platinum, cobalt levels from 0.10 to 19.29 μg/kg, copper levels from 0.30 to 976.14 μg/kg, and iron levels from 0.06 to 53.08 mg/kg. This study established a robust analytical approach for evaluating the heavy metal elution from oxygen absorbers, ensuring safety in the food industry.

## 1. Introduction

Smart packaging is classified into active packaging, which interacts with food and absorbs or releases substances into the surrounding environment, and intelligent packaging, which monitors environmental changes inside and outside the packaging [1]. Active packaging is defined as packaging that can extend the shelf life while maintaining food quality [2], and the main types used are oxygen absorbers, carbon dioxide absorbers/emitters, flavor-releasing/absorbing systems, and ethylene absorbers. Intelligent packaging enables consumers to monitor the status of packaged foods during storage and delivery, supports consumers in purchasing and consuming food, and improves product quality by alerting them to potential problems [3].

There are various types of oxygen absorbers, including iron-based, platinum group metals-based, unsaturated hydrocarbon-based, α-Tocopherol-based, ascorbic acid-based, and enzyme-based oxygen absorbers [4]. Oxygen absorbers based on iron utilize iron powder as their core component and employ transition metals to enhance the oxidation–reduction reaction of iron. Copper and cobalt are the primary transition metals used in this context, which is one of the reasons these substances were chosen for analysis. Additionally, there are oxygen absorbers that use platinum and palladium to convert hydrogen and oxygen into water. In the case of platinum-based oxygen absorbers, catalysts like palladium and sodium are employed to facilitate the reaction [5]. Due to the use of platinum in such applications, platinum was also included as a subject of analysis. In oxygen absorbers based on ascorbic acid, iron and copper are also used as catalysts to facilitate the reaction. Enzyme-based oxygen absorbers utilize copper-containing oxidative enzymes obtained from copper, such as copper-containing oxidoreductases. This widespread use of heavy metals like iron, copper, cobalt, and platinum in most oxygen absorbers suggests the potential for migration into food products [4].

The presence of oxygen in food packaging accelerates the oxidation of food, promoting quality changes such as the occurrence of forehead and odor, browning, and nutritional loss. The most frequently used active packaging technology to remove oxygen from food is the oxygen absorber, which minimizes changes in food quality owing to oxidation by absorbing the remaining oxygen in the food package [2]. Most oxygen absorbers are based on iron powder, which reacts with water in food to absorb oxygen in food packages through redox reactions [4,6]. Iron removes oxygen by forming iron oxide through a reaction with it (Fe + O_2_ → Fe_x_O_y_) [7]. Iron in the oxygen absorber has a greater affinity for oxygen than food; therefore, iron and oxygen react first, removing oxygen and reducing the oxidation of food. Certain metals, such as copper and cobalt, are commonly used as catalysts to enhance this oxidation–reduction reaction. Transition metals like copper and cobalt are not only employed in oxygen absorbers based on iron but also in those based on substances like ascorbic acid and enzymes. Consequently, the likelihood of their presence in packaging materials, along with iron, is high, suggesting a significant potential for migration into food products. In the case of platinum, it is used alongside palladium in environments containing hydrogen within food packaging. They function by converting hydrogen and oxygen into water, thereby removing oxygen, with sodium and palladium serving as catalysts to facilitate the reaction. Therefore, platinum is also used in oxygen absorbers and is considered to have the potential to migrate into food [4].

Copper, cobalt, platinum, and iron are analytical heavy metals. These substances are not life-threatening harmful substances, but their accumulation in the body can harm health. According to the National Institutes of Health (NIH), an excessive intake of iron causes gastritis, gastric lesions, and organ failure. When iron accumulates in the body, it can lead to either acute or chronic toxicity. Iron can generate free radicals, potentially causing cancer because of iron intoxication. Furthermore, iron overload can lead to a reduced influx of Ca^2+^ into cells, which, in turn, decreases neurotransmitter release and impairs signal transmission, possibly giving rise to neurological disorders [8]. The maximum allowable intake of iron is 45 mg/day for 19+ years old individuals. Chronic exposure to copper causes liver damage and gastrointestinal disorders, according to the NIH. Copper is non-biodegradable, and when it deposits in the brain, it can lead to neurotoxicity known as “Wilson’s disease”. It also affects the kidneys, causing kidney dysfunction. According to the International Agency for Research on Cancer (IARC), cobalt is classified as Group 2A, indicating that it is a possible carcinogen for humans [9]. The maximum allowable copper intake is 10,000 mcg/day for 19+ years old individuals. An excessive intake of cobalt increases the risk of heart muscle disease and heart disease. In addition, it damages pancreatic alpha cells and causes hyperglycemia [10]. Cobalt can induce oxidative stress and the generation of reactive oxygen species, leading to damage and toxicity in the liver, kidney, pancreas, and heart tissues. When iron accumulates in the body, it can lead to either acute or chronic toxicity. Iron can generate free radicals, potentially causing cancer because of iron intoxication [8,9]. Platinum intake causes local liver cell necrosis and liver damage [11]. Platinum and platinum compounds can potentially cause kidney damage and impair kidney function. Additionally, they can lead to damage to the stomach and mucous membranes, resulting in symptoms such as nausea and vomiting, indicating gastrointestinal toxicity [12]. Iron, copper, cobalt, and platinum were chosen as subjects of analysis because they possess toxicity, which can be harmful to the human body.

Food simulants are reagents with properties similar to those of food and they can be used instead. Each country has different types of food simulant requirements according to the characteristics of each food. In the US, 10% ethanol is used for aqueous and acidic foods, 10% and 50% ethanol for alcoholic foods, and cooking oil is used as a food simulant for fatty foods [13]. In the EU, 10% ethanol is used for hydrophilic food and 3% acetic acid is used for hydrophilic food with a pH < 4.5. More lipophilic alcoholic food requires 20% ethanol, 50% ethanol is used for lipophilic/alcoholic food (oil in water emulsion), and lipophilic food requires vegetable oil as a food simulant [14]. Table 1 presents the food simulants stipulated by the Ministry of Food and Drug Safety in Korea, which include n-heptane, 20% ethanol, 50% ethanol, water, and 4% acetic acid. These food simulants represent the following food groups. Oil and fatty foods with an oil content of 20% or more correspond to n-heptane, those with an alcohol content of 20% or less correspond to 20% ethanol, and those with an alcohol content of 20% or more correspond to 50% ethanol. Foods with pH 5 or less correspond to 4% acetic acid, and those with pH 5 or more correspond to water. These five food simulants were used in this experiment [15].

The aim of this study was to develop and evaluate an analysis method of heavy metals transferred from oxygen absorbers to food simulants and to monitor the amount of heavy metals transferred to oxygen absorbers used at home and abroad. Inductively coupled plasma-mass spectrometry (ICP-MS) was used for precise and accurate measurements of heavy metals (cobalt, copper, platinum, and iron). According to the Korean Ministry of Food and Drug Safety, there are established methods and permissible limits for analyzing heavy metals present in food. However, there is a lack of regulation and standards for heavy metals that may migrate from packaging materials into food simulants. Given that food packaging comes into direct contact with food items, there is a need for standards regarding the quantity of heavy metals that may migrate from packaging into the food simulants. This study was conducted to develop a method to analyze the amount of heavy metals migrating from packaging to food simulants, quantifying the migration levels and establishing the foundation and basis for regulatory standardization.

## 2. Materials and Methods

### 2.1. Reagents and Materials

Deionized water was purified using an Fpwps501 Ultrapure Distilled Water Purification System from Human Science (Hanam, Republic of Korea). Ultrapure nitric acid (HNO_3_, 70%) was purchased from Duksan (Ansan, Republic of Korea). Nylon membrane filters (0.45 µm) were obtained from Advantec (Chiyoda City, Japan). The mixed standard solution for the calibration curves for Cu and Co, IV-IVPMS-71A 10 μg/mL multi-element, was obtained from Inorganic Ventures (Christiansburg, VA, USA). The standard solution for the iron calibration curve, Iron standard for ICP 1000 mg/L, was obtained from Sigma-Aldrich (Saint Louis, MO, USA). The single standard solution for the platinum calibration curve, Platinum standard solution 1000 mg/L, was obtained from Kanto Chemical (Tokyo, Japan). All standard solutions were prepared by serial dilution using a 3% HNO_3_ concentration.

### 2.2. Instrumentation

ICP-MS was performed with a quadrupole iCAP-RQ equipped with a nebulizer, Teflon spray chamber, nickel sampling cone, and nickel skimmer cone (Thermo Fisher Scientific, Waltham, MA, USA). The sample solution was injected via a peristaltic pump from a tube attached to an autosampler ASX-560 (ThermoFisher Scientific, Waltham, MA, USA). Liquid argon and helium (99.99%) gases were used. Before each experiment, the instrument was calibrated for daily operation using iCAP Q/Qnova TUNE Solution (ThermoFisher Scientific (1323770) USA). The operating conditions are listed in Table 2.

Sample digestion was performed using the microwave-assisted acid digestion system Terminal 640 (Milestone, Bergamo, Lombardia, Italy), which consisted of a rotor for 10 MKM023 HPV-100 TFM Vessels and operated via a method-sample-run process. The operating conditions involved a gradual increase in temperature to 80 °C at 1000 W over 5 min, followed by a 5 min drying phase at 90 °C with the same power setting. Subsequently, the temperature was raised to 190 °C and maintained at 1000 W for 15 min. It was then maintained for 20 min under the same conditions.

### 2.3. Sample Preparation (Food Simulants)

According to the test methods for food utensils, containers, and packaging materials, five food simulant types were selected for each type of food. Water was used as a food simulant for foods with a pH above 5. Acetic acid (≥99.9%) was supplied by Sigma-Aldrich (Saint Louis, MO, USA) and subsequently diluted to 4%. A solution of 4% acetic acid was used in foods with pH 5 or below. For alcoholic beverages, ethanol (anhydrous, 99%) obtained from Samchun (Pyeongtaek, Republic of Korea) was diluted to 20% and 50%. n-Heptane (HPLC grade, 99%) was obtained from Daejung (Siheung, Republic of Korea) as a food simulant for oily and fatty food types.

### 2.4. Sample Digestion and Pre-Processing

To develop a method for analyzing heavy metals transferred from oxygen absorbers to food simulants, a test solution for elution testing was first prepared and microwave pre-treatment was conducted.

#### 2.4.1. Elution Test

The elution test was conducted as part of the preparation of the elution test solution for each material by Article 2, Paragraph 6 of the Ministry of Food and Drug Safety Notice, ‘Food Utensils, Containers, and Packaging Code’ [15]. The test solution was prepared and employed following the protocols specified for the preparation of synthetic resin elution test solutions.

The cell was filled with the extraction solution at a ratio of 2 mL per cm^2^ of the cross-sectional area of the oxygen absorber, heated to 70 °C, and then the sample was placed with the surface in contact with the food facing downwards. A Teflon gasket was placed on top, and the lid was tightly clamped to prevent leakage of the solvent.

The cell was then inverted, and the oxygen absorber was brought into contact with the extraction solution (water, 4% acetic acid, 20%, and 50% ethanol) before being left to stand in a dry oven at 70 °C for 30 min to obtain the test solution. When using n-heptane as the extraction solvent, the sample was left to stand for 1 h at 25 °C and used as the test solution.

#### 2.4.2. Microwave Digestion

A total of 0.5 g (±0.0050 g) of the extracted food simulant was placed in a microwave vessel with 1 mL of ultrapure water. After 5 min of pre-oxidation, 8 mL of 70% HNO_3_ was added and pre-oxidized for 30 min. The sample was then pre-treated using a microwave. After pre-treatment, the digested samples were diluted in 50 mL of ultrapure water. The resulting solutions were filtered through 0.45 µm nylon membrane filters for ICP-MS analysis.

### 2.5. Method Validation

The ICP-MS analytical method was validated for linearity (R^2^), limits of detection (LOD), limits of quantification (LOQ), accuracy (%), and precision (%) on food simulant samples. The linearity of the ICP-MS method was evaluated using the least square linear regression. The calibration curves for each element were analyzed, and the slope, intercept, and coefficient of determination were calculated [16,17]. The LOD and LOQ values were determined as the concentrations that corresponded to 3 and 10 times, respectively, the standard deviation of the blank solution measurements divided by the slope of the calibration curves (Equation (1)).
LOD = 3.3 × σ/S, LOQ = 10 × σ/S(1)

The accuracy and precision of the method were evaluated by analyzing diluted standard solutions that had been spiked. The calibration curves for all the analytes were generated using five different concentrations. The analysis of Cu and Co used standard mixtures at 0.50, 1.00, 5.00, 10.00, and 20.00 µg/L. The analysis of iron used standard solutions at 1.00, 2.00, 5.00, 10.00, and 20.00 mg/L. The analysis of platinum used standard solutions at 0.50, 1.00, 5.00, 10.00, and 20.00 µg/L. These validation parameters followed CODEX guidelines [18]. For accuracy estimations, the calibration curve was evaluated in five different concentrations and three repetitions for intra-day and inter-day measurements. Accuracy was calculated using Equation (2).
Accuracy (%) = (Cmean − Cblk)/Cspiked × 100 (%)(2)

The average concentration of the standard, blank, and spiked samples are denoted as Cmean, Cblk, and Cspiked, respectively. Precision was determined by calculating the coefficient of variation (CV) using Equation (3).
CV (%) = Csd/Cmean × 100 (%)(3)

The standard deviation of the concentration is denoted as Csd.

### 2.6. Statistical Analysis

All analyses were performed in triplicate, and the data are presented as mean ± standard deviation (SD).

## 3. Results and Discussion

### 3.1. ICP-MS Method Validation

The calibration curves were generated for the four heavy metals at five different concentrations (0.5, 1.0, 5.0, 10.0, and 20.0 µg/kg for Co, Cu, and Pt and 0.5, 1.0, 5.0, 10.0, and 20.0 mg/kg for Fe). The linearity of the curves is expressed as the coefficient of determination (R^2^), which was greater than 0.99, indicating excellent linearity within the concentration range tested. For the water, 4% acetic acid, 20% ethanol, 50% ethanol, and n-heptane matrices, the LOQ ranged from 0.07 to 663.6 µg/kg and the LOD ranged from 0.0 to 21.9 µg/kg. The LOQ, LOD, and linearity of the four heavy metals are presented in Table 3.

The accuracy values were utilized to indicate the heavy metal recoveries, which were 95.03% to 101.86% for water, 4% acetic acid, 20% ethanol, 50% ethanol, and n-heptane matrices. The accuracy and precision of the heavy metal analysis in water, 4% acetic acid, 20% ethanol, 50% ethanol, and n-heptane matrices were in the range of 84.37–112.09% and 0.13–2.74% (intraday), and 85.13–113.28% and 0.20–11.51% (interday). The corresponding data for the four heavy metals are presented in Table 4. Interday and intraday accuracy satisfied the 80–120% range set by CODEX, and the precision was also within 15%, showing that the test method was valid.

### 3.2. Cross-Validation of ICP-MS

To establish an objective test method, cross-validation was performed using a matrix spiked with three concentrations of metals: low, medium, and high. The pre-treated 4% acetic acid eluate was used as the same sample. The cross-validation was conducted at Dongguk University Biomedical Campus, Dongguk University Seoul Campus Public Instrument Center, and Korea Health Food Association. Dongguk University Seoul Campus Public Instrument Center and used a Perkin Elmer’s ELAN DRC-e ICP-MS; the Korea Health Food Association used an Agilent’s 7850 ICP-MS. The accuracy of the heavy metal analysis was observed to be 96.2–109.33% during the cross-validation, and the analysis levels are presented in Table 5.

### 3.3. Analysis of the Elution Test

Elution experiments into food were performed with 20 types of oxygen absorbers (8 types in Korea and 12 types in other countries) used both domestically and internationally. The experiments were carried out following the protocol mentioned above, and the test solutions were subjected to microwave pre-treatment and subsequently analyzed using ICP-MS.

Before contacting the oxygen absorbers with food simulants, the amounts of iron, cobalt, copper, and platinum present in the simulants were determined through the same pre-processing method. The quantities of heavy metals in the food simulants were analyzed using ICP-MS, and the results are presented in Table 6. Cobalt exhibited values ranging from 0.204 to 0.248 μg/kg, copper ranged from 0.661 to 1.300 μg/kg, platinum ranged from 0.005 to 0.007 μg/kg, and iron ranged from 0.053 to 0.058 mg/kg. The analysis results show that all four heavy metals were most abundant in n-heptane. Subsequently, the heavy metal content in the oxygen absorbers was determined by subtracting the amounts found in the food simulant.

In the elution test conducted with water as a food simulant, iron was detected in the concentration range of 0.06 to 8.90 mg/kg, cobalt ranged from 1.20 to 8.60 μg/kg, and copper exhibited concentrations ranging from 0.30 to 209.60 μg/kg. In contrast, platinum did not exhibit any detectable elution. These results are shown in Table 7.

In the elution test performed using a food simulant of 4% acetic acid, elutions in the range of 0.07 to 53.08 mg/kg of iron, 0.20 to 5.20 μg/kg of cobalt, and 2.80 to 976.14 μg/kg of copper were observed. However, no elution of platinum was observed. The outcomes are presented in Table 8. In the iron and copper elutions, the highest amounts were detected when eluted with 4% acetic acid, which is a food simulant corresponding to foods other than mainstream and fatty foods and with a pH not exceeding 5. Meat and fish products, mainly used with oxygen absorbers, have a pH of about 3 to 5.5 [19]. Therefore, heavy metals are highly likely to be transferred to foods where oxygen absorbers are mainly used.

In the elution test performed using a food simulant of n-heptane, a range of 0.06 to 32.46 mg/kg of iron, 1.39 to 19.29 μg/kg of cobalt, and 1.20 to 176.73 μg/kg of copper elutions were observed. Moreover, platinum eluted at 0.10 μg/kg in oxygen absorbers for long-term food storage. These results are shown in Table 9. Platinum elution as a food simulant could not be found in almost all oxygen absorbers, but it was the only one detected in n-heptane. The results of the platinum transition show that this metal is rarely used in oxygen absorbers, or that it is likely to be transferred to fatty foods.

In the elution test performed using a food simulant of 20% ethanol, elutions in the range of 0.09 to 10.14 mg/kg of iron, 0.10 to 14.71 μg/kg of cobalt, and 2.83 to 89.98 μg/kg of copper were observed. In contrast, no platinum elution was observed. These results are shown in Table 10. A large amount of iron and copper was also eluted in 20% ethanol, a representative simulant for alcohols with less than 20%. However, most liquids in bottles, such as alcoholic beverages, are sealed using a crown cap-type lid instead of a sachet-type oxygen absorber to remove oxygen [7]. Therefore, we conclude that heavy metals are transferred from oxygen absorbers and are less likely to affect human health.

Table 11 presents the elution test performed with 50% ethanol as a food simulant, which yielded eluted concentrations of iron ranging from 0.07 to 5.33 mg/kg, cobalt ranging from 0.89 to 10.09 μg/kg, and copper ranging from 0.30 to 66.31 μg/kg. No platinum elution was observed during the experiments.

According to Mi-Kyung Jang, the amount of heavy metals dissolved into the simulant is affected by changes in temperature, with increased dissolution occurring as the temperature increases [20]. In the present study, the experiments were performed at 70 °C for 30 min. These experiments should also be performed under actual temperature, refrigeration, and freeze conditions to compare the different elution amounts. In addition, the transfer of heavy metals from food packaging to food may decrease when food is washed [21]. In the case of a food simulant, washing is not possible. Therefore, the amount of heavy metal transfer during cleaning and at the micro-scale could not be compared. It is expected that more detailed regulations of the standards and methods of using oxygen absorbers are possible if these absorbers are tested with food samples instead of food simulants.

The EU and the US strictly regulate substances that contact food. The EU stipulates the materials that contact food “to be manufactured by good manufacturing practices so that ingredients are not transferred to food in the following amounts under regular or predictable conditions of use”. Currently, there are standards for heavy metal detection in food, but there are no standards for heavy metals transferred from food packaging to food. In addition, in Korea, food packaging materials are regulated for finished products rather than for the prior approval of materials. Therefore, there is no regulation of the substances used. For these reasons, the regulation and implementation of food packaging materials should be created and managed in accordance with accepted standards.

## 4. Conclusions

Heavy metals in packaging materials can be transferred to food or food simulants. Heavy metals (cobalt, copper, platinum, and iron) can be harmful if they accumulate excessively in the body. In this study, the elution of heavy metals from oxygen absorbers to food simulants (water, 4% acetic acid, 20% ethanol, 50% ethanol, and n-heptane) was analyzed and monitored using a proven method. The analytical method of ICP-MS was validated to investigate the amount of heavy metals eluted from the oxygen absorber to the food simulant.

This study provides important information for analyzing heavy metals eluted when oxygen absorbents are brought into contact with food simulants. The analysis method was used for heavy metals such as cobalt, copper, platinum, and iron in the five food simulants. All R^2^ values representing linearity met the range of 0.995 or higher. The accuracy was within the range of 80–120%, and the precision was within 15%. As a result of verifying the developed analysis method, the method was validated by satisfying the validation guidelines stipulated by CODEX. Monitoring was conducted using the developed analysis method. In the five simulants, 0.06–53.08 mg/kg of iron, 1.20–19.29 μg/kg of cobalt, 0.30–976.14 μg/kg of copper, and 0.10 μg/kg of platinum were detected. However, there is no reference standard for heavy metals transferred from food packaging to food and food simulants, and thus it is not known whether the safety levels have been exceeded. Safety standards for materials that can be implemented in packaging materials are therefore required. The results of this study can be used as data on the analysis method of elution from oxygen absorber to food and the amount of heavy metals eluted into food; furthermore, they can also be used to establish elution standards for various active packaging materials, including domestic oxygen absorbers.

## Figures and Tables

**Table 1 foods-12-03686-t001:** Classification of food simulant according to food characteristics.

Type of Food		Food Simulants
Oils and fatty foods (Food containing 20% or more of the total, surface, or part of the surface of the food)		n-Heptane
Alcoholic beverages	Alcoholic beverages containing not more than 20% alcohol	20% ethanol
	Alcoholic beverages containing more than 20% alcohol	50% ethanol
Food other than oils and fatty foods and alcoholic beverages	Food with a pH of 5 or lower	4% acetic acid
	Food exceeding pH 5	Water

**Table 2 foods-12-03686-t002:** Operating conditions of ICP-MS.

Parameter	Value
RF power	1550 W
Ar gas flow rate (mL/min) Sample uptake rate	Auxiliary: 0.8, nebulizer: 1.0 1.0 mL/min
Nebulizer Spray chamber Torch	Concentric type Cyclonic type Demountable
Interface cones	Nickel
Quadrupole chamber Dwell time (ms)	1 × 10^−6^ torr 600
Analytical masses	Fe (57), Co (59), Cu (63), Pt (195)
Analytes and measurement mode	KED

**Table 3 foods-12-03686-t003:** The LOD, LOQ, linearity, equation, and R^2^ of the four heavy metals.

Matrix Type	Heavy Metal Element	LOD ^a^ (mg/kg)	LOQ ^b^ (mg/kg)	Linearity Equation ^c^	R^2^
Water	Co	0.00002	0.00007	y = 37,537x + 7795.5	1.0000
	Cu	0.0001	0.0003	y = 29,064x + 26,736	1.0000
	Pt	0.0002	0.0005	y = 3,982,718x − 229,619	1.0000
	Fe	0.1123	0.3404	y = 330,348x + 23,425	1.0000
4% acetic acid	Co	0.0001	0.0002	y = 37,542x + 7752.8	1.0000
	Cu	0.0000	0.0001	y = 28,891x + 30,671	0.9996
	Pt	0.0001	0.0003	y = 3,971,346x + 55,332	0.9999
	Fe	0.1032	0.3129	y = 345,386x − 31,532	0.9999
20% ethanol	Co	0.00002	0.0001	y = 37,209x + 9605.6	0.9998
	Cu	0.0001	0.0002	y = 29,079x + 28,236	0.9986
	Pt	0.00002	0.0001	y = 4,144,907x − 35,874	1.0000
	Fe	0.2190	0.6636	y = 351,745x + 32,137	0.9998
50% ethanol	Co	0.0001	0.0004	y = 38,615x + 7459.2	1.0000
	Cu	0.0001	0.0002	y = 30,523x + 21,463	1.0000
	Pt	0.00003	0.0001	y = 4,289,917x − 21,839	1.0000
	Fe	0.0440	0.1333	y = 380,607x + 42,364	0.9989
n-Heptane	Co	0.0001	0.0003	y = 40,494x + 9085.2	1.0000
	Cu	0.0001	0.0002	y = 31,664x + 41,842	0.9993
	Pt	0.0001	0.0003	y = 4,388,571x + 118,922	1.0000
	Fe	0.0735	0.2228	y = 427,438x + 39,190	0.9998

^a^ set up in a signal-to-noise ratio (S/N) = 3.3; ^b^ set up in a signal-to-noise ratio (S/N) = 10; ^c^ numbers express mean values (n = 3).

**Table 4 foods-12-03686-t004:** Accuracy and precision (% CV) analysis of the four heavy metals.

Matrix Type	Heavy Metal Element	Intraday (*n* = 3)		Interday (*n* = 3)	
		Accuracy (Mean ± SD)	Precision (Mean ± SD)	Accuracy (Mean ± SD)	Precision (Mean ± SD)
Water	Co (μg/kg)	100.68 ± 0.55	0.73 ± 0.48	100.21 ± 0.37	2.13 ± 0.55
	Cu (μg/kg)	100.71 ± 1.01	0.79 ± 0.50	99.09 ± 1.49	2.44 ± 1.28
	Pt (μg/kg)	99.68 ± 2.44	0.94 ± 0.24	100.07 ± 2.45	2.19 ± 0.96
	Fe (mg/kg)	99.57 ± 1.24	1.33 ± 0.41	99.77 ± 1.17	10.37 ± 1.24
4% acetic acid	Co (μg/kg)	99.69 ± 0.95	0.63 ± 0.27	100.11 ± 0.59	1.15 ± 0.46
	Cu (μg/kg)	96.72 ± 6.62	0.87 ± 0.69	97.33 ± 6.03	1.59 ± 0.38
	Pt (μg/kg)	100.62 ± 1.99	0.79 ± 0.51	100.73 ± 2.06	2.22 ± 1.05
	Fe (mg/kg)	96.04 ± 2.99	0.87 ± 0.38	98.45 ± 1.26	8.62 ± 0.96
20% ethanol	Co (μg/kg)	101.17 ± 1.48	0.66 ± 0.5	100.95 ± 1.06	1.35 ± 0.52
	Cu (μg/kg)	99.76 ± 7.98	0.77 ± 0.54	99.98 ± 8.5	1.08 ± 0.42
	Pt (μg/kg)	100.14 ± 2.69	0.65 ± 0.23	100.75 ± 2.54	0.72 ± 0.4
	Fe (mg/kg)	100.35 ± 1.18	0.91 ± 0.52	101.52 ± 0.94	7.29 ± 0.75
50% etnaol	Co (μg/kg)	100.55 ± 0.49	0.82 ± 0.33	100.31 ± 0.57	1.91 ± 0.76
	Cu (μg/kg)	99.76 ± 0.93	0.97 ± 0.89	99.44 ± 1.32	2.12 ± 1.07
	Pt (μg/kg)	100.58 ± 2.68	0.85 ± 0.46	100.76 ± 2.78	0.68 ± 0.3
	Fe (mg/kg)	95.52 ± 7.32	0.61 ± 0.35	96.52 ± 6.32	5.82 ± 1.07
n-Heptane	Co (μg/kg)	101.62 ± 1.91	0.59 ± 0.47	101.45 ± 1.68	1.56 ± 0.93
	Cu (μg/kg)	95.14 ± 6.44	0.95 ± 1.02	95.03 ± 6.88	2.58 ± 2.46
	Pt (μg/kg)	101.98 ± 3.24	0.81 ± 0.35	101.62 ± 3.13	1.07 ± 0.52
	Fe (mg/kg)	100.03 ± 0.40	0.58 ± 0.24	100.51 ± 0.51	3.6 ± 0.28

**Table 5 foods-12-03686-t005:** Cross-validation results of the four heavy metals in 4% acetic acid.

Matrix Type	Heavy Metal Element Concentration	Spike	A ^a^ Mean Value	B ^b^ Mean Value	C ^c^ Mean Value	Accuracy (%)
4% Acetic acid	Co (μg/kg)	1	1.10	1.19	0.98	109.07
		5	5.52	5.74	4.99	108.31
		20	21.35	22.46	19.83	106.05
	Cu (μg/kg)	1	1.09	1.21	0.99	109.33
		5	5.32	4.75	4.94	104.73
		20	20.78	19.86	19.82	103.63
	Pt (μg/kg)	1	1.02	1.00	0.95	98.87
		5	4.88	4.75	4.80	96.20
		20	19.71	19.86	19.11	97.79
	Fe (mg/kg)	1	1.09	1.11	1.00	106.67
		5	5.27	5.38	4.87	103.45
		20	21.59	22.22	19.48	105.48

A ^a^: Dongguk University Biomedical Campus, B ^b^: Dongguk University Seoul Campus Public Instrument Center, C ^c^: Korea Health Food Association.

**Table 6 foods-12-03686-t006:** The amount of heavy metals present in the food simulant.

Heavy Metal Element Concentration	Matrix Type				
	Water	4% Acetic Acid	n-Heptane	20% Ethanol	50% Ethanol
Co (μg/kg)	0.204	0.211	0.248	0.211	0.227
Cu (μg/kg)	0.661	1.013	1.300	0.938	0.877
Pt (μg/kg)	0.007	0.005	0.005	0.005	0.005
Fe (mg/kg)	0.053	0.054	0.058	0.056	0.056

**Table 7 foods-12-03686-t007:** Content of four heavy metals from oxygen absorbers using elution of food simulant (water).

Oxygen Absorbers	Fe (mg/kg)	Co (μg/kg)	Cu (μg/kg)	Pt (μg/kg)
Oxygen Absorber 1	0.40 ± 0.00	2.79 ± 0.00	15.62 ± 0.01	ND ^a^
Oxygen Absorber 2	0.30 ± 0.00	2.58 ± 0.00	13.88 ± 0.00	ND
Oxygen Absorber 3	0.30 ± 0.00	1.91 ± 0.00	0.30 ± 0.00	ND
Oxygen Absorber 4	0.30 ± 0.00	1.40 ± 0.00	ND	ND
Oxygen Absorber 5	0.40 ± 0.00	1.49 ± 0.00	ND	ND
Oxygen Absorber 6	0.30 ± 0.00	1.51 ± 0.00	4.52 ± 0.00	ND
Oxygen Absorber 7	4.42 ± 0.00	6.63 ± 0.00	6.83 ± 0.00	ND
Oxygen Absorber 8	6.56 ± 0.00	4.84 ± 0.00	209.60 ± 0.01	ND
Oxygen Absorber 9	0.50 ± 0.00	1.81 ± 0.00	10.57 ± 0.00	ND
Oxygen Absorber 10	1.30 ± 0.00	2.69 ± 0.00	17.56 ± 0.01	ND
Oxygen Absorber 11	0.40 ± 0.00	1.20 ± 0.00	15.41 ± 0.01	ND
Oxygen Absorber 12	0.60 ± 0.00	1.20 ± 0.00	ND	ND
Oxygen Absorber 13	0.50 ± 0.00	1.30 ± 0.00	ND	ND
Oxygen Absorber 14	1.61 ± 0.00	3.32 ± 0.00	19.11 ± 0.00	ND
Oxygen Absorber 15	0.06 ± 0.00	3.31 ± 0.00	2.90	ND
Oxygen Absorber 16	1.39 ± 0.00	1.79 ± 0.00	3.27 ± 0.01	ND
Oxygen Absorber 17	1.71 ± 0.00	2.31 ± 0.00	ND	ND
Oxygen Absorber 18	8.90 ± 0.00	8.60 ± 0.00	9.50 ± 0.01	ND
Oxygen Absorber 19	2.31 ± 0.00	1.71 ± 0.00	ND	ND
Oxygen Absorber 20	2.42 ± 0.00	2.83 ± 0.00	26.54 ± 0.01	ND

^a^ ND = not detected, the lower limit of detection.

**Table 8 foods-12-03686-t008:** Content of four heavy metals from oxygen absorbers using elution of food simulant (4% acetic acid).

Oxygen Absorbers	Fe (mg/kg)	Co (μg/kg)	Cu (μg/kg)	Pt (μg/kg)
Oxygen Absorber 1	0.50 ± 0.00	0.69 ± 0.00	62.30 ± 0.00	ND
Oxygen Absorber 2	1.49 ± 0.00	1.59 ± 0.00	170.87 ± 0.00	ND
Oxygen Absorber 3	ND	ND	41.93 ± 0.00	ND
Oxygen Absorber 4	0.10 ± 0.00	0.70 ± 0.00	539.25 ± 0.03	ND
Oxygen Absorber 5	ND	ND	976.14 ± 0.04	ND
Oxygen Absorber 6	0.20 ± 0.00	ND	283.82 ± 0.01	ND
Oxygen Absorber 7	0.59 ± 0.00	0.50 ± 0.00	ND	ND
Oxygen Absorber 8	1.81 ± 0.00	1.91 ± 0.00	9.63 ± 0.01	ND
Oxygen Absorber 9	0.20 ± 0.00	ND	343.10 ± 0.02	ND
Oxygen Absorber 10	0.69 ± 0.00	1.69 ± 0.00	6.75 ± 0.01	ND
Oxygen Absorber 11	1.59 ± 0.00	1.79 ± 0.00	11.52 ± 0.00	ND
Oxygen Absorber 12	53.08 ± 0.00	5.16 ± 0.00	9.13 ± 0.00	ND
Oxygen Absorber 13	1.50 ± 0.00	2.90 ± 0.00	41.63 ± 0.01	ND
Oxygen Absorber 14	1.00 ± 0.00	5.20 ± 0.00	112.02 ± 0.01	ND
Oxygen Absorber 15	0.07 ± 0.00	2.72 ± 0.00	31.72 ± 0.00	ND
Oxygen Absorber 16	3.40 ± 0.00	1.00 ± 0.01	15.11 ± 0.01	ND
Oxygen Absorber 17	4.09 ± 0.00	0.70 ± 0.00	86.01 ± 0.01	ND
Oxygen Absorber 18	0.60 ± 0.00	0.20 ± 0.00	2.80 ± 0.01	ND
Oxygen Absorber 19	0.40 ± 0.00	0.20 ± 0.00	ND	ND
Oxygen Absorber 20	1.70 ± 0.00	0.30 ± 0.00	6.31 ± 0.01	ND

**Table 9 foods-12-03686-t009:** Content of four heavy metals from oxygen absorbers using elution of food simulant (n-heptane).

Oxygen Absorbers	Fe (mg/kg)	Co (μg/kg)	Cu (μg/kg)	Pt (μg/kg)
Oxygen Absorber 1	1.10 ± 0.00	1.40 ± 0.00	3.60 ± 0.01	ND
Oxygen Absorber 2	2.72 ± 0.00	4.03 ± 0.00	10.38 ± 0.01	ND
Oxygen Absorber 3	1.20 ± 0.00	2.00 ± 0.00	1.20 ± 0.00	ND
Oxygen Absorber 4	1.20 ± 0.00	2.20 ± 0.00	5.01 ± 0.00	ND
Oxygen Absorber 5	4.81 ± 0.00	4.51 ± 0.00	57.76 ± 0.02	ND
Oxygen Absorber 6	0.71 ± 0.00	1.92 ± 0.00	ND	ND
Oxygen Absorber 7	0.50 ± 0.00	6.39 ± 0.00	2.10 ± 0.00	ND
Oxygen Absorber 8	1.21 ± 0.00	7.77 ± 0.00	14.02 ± 0.00	ND
Oxygen Absorber 9	2.79 ± 0.00	2.39 ± 0.00	2.69 ± 0.00	ND
Oxygen Absorber 10	0.80 ± 0.00	1.39 ± 0.00	39.21 ± 0.00	ND
Oxygen Absorber 11	1.19 ± 0.00	3.96 ± 0.00	1.49 ± 0.01	ND
Oxygen Absorber 12	1.51 ± 0.00	3.62 ± 0.00	ND	ND
Oxygen Absorber 13	32.46 ± 0.00	5.93 ± 0.00	77.19 ± 0.00	ND
Oxygen Absorber 14	1.90 ± 0.00	6.01 ± 0.00	7.41 ± 0.00	ND
Oxygen Absorber 15	0.06 ± 0.00	ND	ND	ND
Oxygen Absorber 16	0.10 ± 0.00	10.54 ± 0.00	ND	ND
Oxygen Absorber 17	0.90 ± 0.00	13.19 ± 0.01	176.73 ± 0.02	ND
Oxygen Absorber 18	2.12 ± 0.00	19.29 ± 0.00	10.81 ± 0.00	0.10 ± 0.00
Oxygen Absorber 19	1.00 ± 0.00	17.12 ± 0.00	57.17 ± 0.00	ND
Oxygen Absorber 20	0.80 ± 0.00	13.79 ± 0.00	37.88 ± 0.01	ND

**Table 10 foods-12-03686-t010:** Content of four heavy metals from oxygen absorbers using elution of food simulant (20% ethanol).

Oxygen Absorbers	Fe (mg/kg)	Co (μg/kg)	Cu (μg/kg)	Pt (μg/kg)
Oxygen Absorber 1	0.20 ± 0.00	0.10 ± 0.00	46.89 ± 0.00	ND
Oxygen Absorber 2	ND	ND	2.83 ± 0.01	ND
Oxygen Absorber 3	0.30 ± 0.00	ND	5.90 ± 0.00	ND
Oxygen Absorber 4	0.20 ± 0.00	ND	ND	ND
Oxygen Absorber 5	0.10 ± 0.00	ND	77.65 ± 0.01	ND
Oxygen Absorber 6	0.30 ± 0.00	ND	51.04 ± 0.01	ND
Oxygen Absorber 7	1.10 ± 0.00	4.80 ± 0.00	12.50 ± 0.01	ND
Oxygen Absorber 8	0.70 ± 0.00	6.52 ± 0.00	4.21 ± 0.01	ND
Oxygen Absorber 9	0.20 ± 0.00	ND	89.98 ± 0.01	ND
Oxygen Absorber 10	0.30 ± 0.00	2.68 ± 0.00	ND	ND
Oxygen Absorber 11	0.50 ± 0.00	2.58 ± 0.00	ND	ND
Oxygen Absorber 12	0.61 ± 0.00	2.12 ± 0.00	ND	ND
Oxygen Absorber 13	0.59 ± 0.00	3.57 ± 0.00	8.12 ± 0.01	ND
Oxygen Absorber 14	10.14 ± 0.00	14.71 ± 0.00	83.47 ± 0.02	ND
Oxygen Absorber 15	0.09 ± 0.00	7.46 ± 0.00	61.32 ± 0.01	ND
Oxygen Absorber 16	0.70 ± 0.00	5.57 ± 0.00	10.14 ± 0.00	ND
Oxygen Absorber 17	1.11 ± 0.00	5.65 ± 0.00	13.91 ± 0.01	ND
Oxygen Absorber 18	0.60 ± 0.00	4.74 ± 0.00	ND	ND
Oxygen Absorber 19	0.70 ± 0.00	5.94 ± 0.01	ND	ND
Oxygen Absorber 20	1.00 ± 0.00	5.68 ± 0.00	ND	ND

**Table 11 foods-12-03686-t011:** Content of four heavy metals from oxygen absorbers using elution of food simulant (50% ethanol).

Oxygen Absorbers	Fe (mg/kg)	Co (μg/kg)	Cu (μg/kg)	Pt (μg/kg)
Oxygen Absorber 1	ND	ND	4.97 ± 0.01	ND
Oxygen Absorber 2	0.60 ± 0.00	0.89 ± 0.00	10.44 ± 0.00	ND
Oxygen Absorber 3	ND	ND	17.86 ± 0.00	ND
Oxygen Absorber 4	0.60 ± 0.00	ND	4.08 ± 0.01	ND
Oxygen Absorber 5	1.20 ± 0.00	ND	0.30 ± 0.00	ND
Oxygen Absorber 6	0.50 ± 0.00	ND	21.53 ± 0.01	ND
Oxygen Absorber 7	0.71 ± 0.00	6.97 ± 0.00	6.16 ± 0.01	ND
Oxygen Absorber 8	1.00 ± 0.00	6.29 ± 0.00	66.31 ± 0.01	ND
Oxygen Absorber 9	0.20 ± 0.00	0.60 ± 0.00	41.84 ± 0.01	ND
Oxygen Absorber 10	0.71 ± 0.00	5.55 ± 0.00	12.40 ± 0.01	ND
Oxygen Absorber 11	0.30 ± 0.00	4.72 ± 0.00	9.14 ± 0.00	ND
Oxygen Absorber 12	2.70 ± 0.00	8.31 ± 0.00	14.62 ± 0.01	ND
Oxygen Absorber 13	0.80 ± 0.00	6.29 ± 0.00	11.69 ± 0.01	ND
Oxygen Absorber 14	0.99 ± 0.00	6.35 ± 0.00	17.08 ± 0.01	ND
Oxygen Absorber 15	0.07 ± 0.00	5.95 ± 0.00	10.12 ± 0.01	ND
Oxygen Absorber 16	0.40 ± 0.00	5.05 ± 0.00	22.42 ± 0.00	ND
Oxygen Absorber 17	1.11 ± 0.00	5.45 ± 0.00	30.69 ± 0.01	ND
Oxygen Absorber 18	1.41 ± 0.00	4.34 ± 0.00	2.42 ± 0.00	ND
Oxygen Absorber 19	2.93 ± 0.00	10.09 ± 0.00	29.66 ± 0.01	ND
Oxygen Absorber 20	5.33 ± 0.00	7.94 ± 0.00	18.49 ± 0.01	ND

## Data Availability

The data used to support the findings of this study can be made available by the corresponding author upon request.

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
