# Peer review of "Evaluation of Oxygen Absorbers Using Food Simulants and Inductively Coupled Mass Spectrometry"

_foods, 2023, doi:10.3390/foods12193686_

Round 1

Reviewer 1 Report

Comments and Suggestions for Authors

·         The authors have chosen to analyze cobalt, copper, platinum, and iron. However, they do not provide an explanation for their selection of these four metals. The only explanation provided in lines 46-47 is quite inadequate and lacks accompanying references. An acceptable explanation is given only for iron, leaving the rationales for the other metals notably insufficient. Merely stating that they have potential health risks is not satisfactory; there should be a discussion as to why one would anticipate finding these metals in the packaging.

·         In general, the introduction is deficient as it fails to delineate existing knowledge in the literature and the authors' proposed contributions in relation to the current state of knowledge. Consequently, the potential novelty of the work and its relevance remain unaddressed.

·         In a validation study, determining linearity through the calculation of R-squared is inadequate. R-squared indicates how well the experimental data points fit. To assess linearity, specific tests such as the Mandel test should be conducted.

·         The description of the elution test lacks clarity.

·         However, the principal issue with this work lies in the proposed approach for method validation. All validation studies employ extraction solutions spiked with varying amounts of individual metals. No tests are conducted on real samples, such as repeatability on real samples. Even the recovery calculations solely based on the extraction solution do not offer a genuine evaluation of the extraction solution's efficacy. In my opinion, there should be at least one test involving the measurement of the actual quantities present in the sample (packaging) before and after extraction. Only then can the authors ascertain the effectiveness of the extraction solution. The authors appear to have merely adhered to guidelines, yet method validation entails a more comprehensive process. It should be meticulously planned (in accordance with guidelines) and tailored to the nature of the experiment, involving real samples for reference. I believe the approach taken by the authors is inadequate. Hence, I suggest that the work needs to be restructured, and in its current form, I do not believe it can be accepted

Reviewer 2 Report

Comments and Suggestions for Authors

I have some comments to the manuscript:

 1. What exactly means " analytical heavy metals"? What about their toxicity?

2.In assessing analytical methods two more factors should be considered: signal drift and interfering effects. Please, comment!

3. Were the conditions for analytical measurements optimized and if yes, how - by experimental design or by single-at-a -time experiments?

Comments on the Quality of English Language

Minor corrections of English is needed

Reviewer 3 Report

Comments and Suggestions for Authors

This work describes the development and validation of an analytical method based on Inductively Coupled Mass Spectrometry, to analyse the elution of heavy metals from oxygen absorbers in contact with food. Different food simulants were used to carry out the study.

I think that the subject is of interest and the results and discussion is coherent. However, I think that the work does not include any information about the oxygen absorbers used.  In addition in the Introduction the authors explain that most oxygen absorbers are based in iron. I think that examples of different absorbers that can produce elution of the heavy metals analysed should be described in the introduction. Therefore, major revision is needed.

Round 2

Reviewer 1 Report

Comments and Suggestions for Authors

Although the work has improved,   I continue to believe that for a study of high scientific interest, the authors should have tested the release of metals by measuring the quantities of metals potentially released from the packaging, following the following steps:

  1. Measure the levels of metals in the packaging.
  2. Measure the levels in the solution before and after contact with the packaging.

At this point, the validation of the analytical data would have made sense. As far as I'm concerned, the validation of only the metal contents in the extracting solution is not of interest. However, this is a personal opinion. 

            In addition  I am aware that R-squared is commonly accepted; however, it is important to highlight its limitations in a validation study. I would like to reiterate that the definition of R-squared is as follows: 'R-squared shows how well the experimental data fit the regression model.' Of course, can't contesting its usage; my comment was simply a suggestion /reflession

Author Response

Thank you for your review.

Reviewer 3 Report

Comments and Suggestions for Authors

In my opinion the authors have correctly adressed the remarks given by the reviewers and the manuscript can be published as it is.

Author Response

We deeply appreciate your valuable comments and reviews. Your insights and advice were very helpful in improving my thesis. Thanks to this, I can now communicate my research more clearly.